# AceReason-Nemotron 1.1:
# Advancing Math and Code Reasoning through SFT and RL Synergy

**Zihan Liu**[1][†] [*]**, Zhuolin Yang**[1]**, Yang Chen**[1]**, Chankyu Lee**[1]**, Mohammad Shoeybi**[1]**,
Bryan Catanzaro**[1]**, Wei Ping**[1][†][‡]

## Abstract

In this work, we investigate the synergy between supervised fine-tuning (SFT) and reinforcement learning (RL) in developing strong reasoning models. We begin by curating the SFT training data through two scaling strategies: increasing the number of collected prompts and the number of generated responses per prompt. Both approaches yield notable improvements in reasoning performance, with scaling the number of prompts resulting in more substantial gains. We then explore the following questions regarding the synergy between SFT and RL: (i) Does a stronger SFT model consistently lead to better final performance after large-scale RL training? (ii) How can we determine an appropriate sampling temperature during RL training to effectively balance exploration and exploitation for a given SFT initialization? Our findings suggest that (i) holds true, provided effective RL training is conducted, particularly when the sampling temperature is carefully chosen to maintain the temperature-adjusted entropy around 0.3, a setting that strikes a good balance between exploration and exploitation. Notably, the performance gap between initial SFT models narrows significantly throughout the RL process. Built on a strong SFT foundation and SFT–RL synergy, our AceReason-Nemotron-1.1 7B model significantly outperforms AceReason-Nemotron-1.0 and achieves new state-of-the-art performance among Qwen2.5-7B-based reasoning models on challenging math and code benchmarks, thereby demonstrating the effectiveness of our post-training recipe.

## 1 Introduction

Math and code reasoning with large language models (LLMs) has been an active area of research for years (Cobbe et al., 2021; Chen et al., 2021). Previous work has primarily focused on enhancing short chain-of-thought (CoT) reasoning (Wei et al., 2022), which is typically acquired through pretraining and supervised fine-tuning (SFT) (e.g., Shao et al., 2024; Hui et al., 2024; Yang et al., 2024b; Liu et al., 2024c).

Since the introduction of OpenAI o1 (OpenAI, 2024) and DeepSeek-R1 (Guo et al., 2025; Liu et al., 2024a), long chain-of-thought (CoT) reasoning, which is acquired through large-scale reinforcement learning (RL), has emerged as a key driver of the remarkable progress in the reasoning capabilities of frontier LLMs (Guo et al., 2025; Qwen-Team, 2025; Yang et al., 2025), with reward signals typically provided by rule-based verifiers. Subsequently, much of the follow-up work has focused on distilling these large frontier models into smaller or mid-sized models through synthetic data generation and SFT-only approach (Bercovich et al., 2025; Moshkov et al., 2025; Ahmad et al., 2025; Yang et al., 2025). Several recent efforts have sought to replicate the success of large-scale RL on smaller base or SFT models (i.e., 7B and 14B ) (e.g., Luo et al., 2025a; He et al., 2025a; Wen et al., 2025; Liu et al., 2025), often leveraging DeepSeek-R1-Distill-Qwen2.5 (Chen et al., 2025) as the initialization checkpoints. However, a systematic study of the synergy between SFT and RL has been limited within the research community and is notably absent from the technical reports of frontier models.

AceReason-Nemotron-1.0-7B (Chen et al., 2025) proposed a stage-wise RL method on math-only and code-only prompts, demonstrating both effectiveness and efficiency. In this work, we take a

---

[*]NVIDIA[1]. [†]: Correspondence to: zihanl@nvidia.com, wping@nvidia.com. [‡]: Technical Lead.

step further by integrating supervised fine-tuning (SFT) and reinforcement learning (RL), probing their training dynamics and the synergy between them to provide a holistic perspective on these post-training techniques for building state-of-the-art reasoning models.

Specifically, we make the following contributions:

1. We begin by scaling SFT training through collecting a large number of prompts, increasing the number of generated responses per prompt, and increasing the number of training epochs. We find that: *(i)* Scaling both the number of prompts and the number of generated responses per prompt leads to substantial improvements in reasoning performance on math and code benchmarks, although scaling the number of prompts yields more significant gains. *(ii)* We observe consistent performance gains from the first to the fifth epoch, with improvements plateauing between the fifth and sixth epochs, regardless of the specific SFT blend used. This suggests that a certain degree of "overfitting" actually enhances test accuracy with long CoT generation, likely due to *exposure bias* in autoregressive models.

2. We initiate RL training from various SFT models and make the following key observations: *(i)* Stronger SFT models continue to produce consistently better results after large-scale RL, although the performance gap narrows during RL training. *(ii)* For a given initial SFT model, selecting an appropriate temperature during RL training is crucial for achieving a good balance between exploration and exploitation. We provide a rule of thumb for setting the sampling temperature such that the temperature-adjusted entropy remains around 0.3, which typically leads to effecive RL training.

3. We systematically study the best strategy during RL training when the final answer is not produced within a specific response length budget (e.g., 24K). Whether we should assign a negative reward or mask out the entire sample (i.e., overlong filtering). Our findings show that overlong filtering provides clear benefits when the token limit is short (e.g., 8K or 16K). However, this advantage diminishes at a 24K token budget, and at 32K, overlong filtering can even degrade model performance.

4. We confirm that the stage-wise RL approach on math-only and code-only prompts remains effective when applied to a range of much stronger SFT models beyond DeepSeek-R1-Distill-Qwen models, demonstrating the broad applicability of this method (Chen et al., 2025). Building on our strong SFT model and insights into the synergy between SFT and RL, our AceReason-Nemotron-1.1 7B model achieves record-high performance among Qwen2.5-7B-based reasoning models on challenging math and code reasoning benchmarks, demonstrating the superiority of our post-training recipe.

## 2 RELATED WORK

The capacity for reasoning is essential to AI and manifests in both the text domain (e.g., Wei et al., 2022; Cobbe et al., 2021; Hui et al., 2024) and the multimodal space (e.g., Dai et al., 2024; Zhu et al., 2025; Ghosh et al., 2025). DeepSeek-R1 (Guo et al., 2025) has demonstrated the effectiveness of verification-based reinforcement learning (RL). This approach has proven particularly useful in domains with structured outputs and well-defined verification criteria (e.g., Chen et al., 2025; Yang et al., 2024b; Liu et al., 2024c; Guo et al., 2025; Luo et al., 2025a). Such rule-based verification eliminates the need for reward modeling and enhances the accuracy of the reward signal.

Several follow-up studies have explored various RL training techniques to improve model performance on math and code reasoning tasks (Luo et al., 2025b; He et al., 2025b; Chen et al., 2025). AceReason-Nemotron-1.0 (Chen et al., 2025) proposed a stage-wise RL method that trains sequentially on math and code prompts, and advocated for strict on-policy GRPO (Shao et al., 2024) training. DAPO (Yu et al., 2025) investigates the use of an overlong penalty—which assigns negative rewards to truncated generations within the response length—in the math domain, and finds that removing it is beneficial. This approach is also adopted by DeepCoder (Luo et al., 2025a). In this work, we conduct a more systematic analysis and find that the overlong penalty should be applied during the later stages of RL training, rather than at the early stages.

Another line of work focuses on supervised fine-tuning (SFT) via distillation from frontier reasoning models trained with reinforcement learning. Notable examples include DeepSeek-R1-Distill-Qwen (Guo et al., 2025), Light-R1 (Wen et al., 2025), OpenMathReasoning (Moshkov et al., 2025),

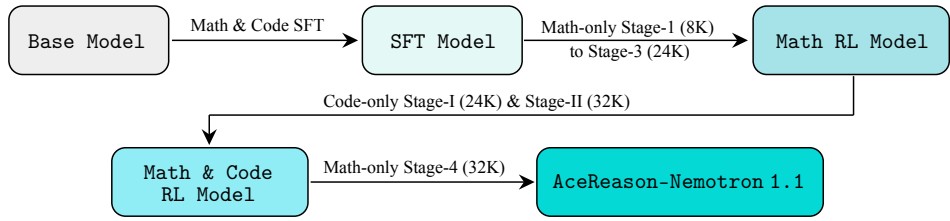

Figure 1: Training Pipeline of AceReason-Nemotron 1.1. We start by performing math and code SFT on a base pretrained model. Next, we conduct three stages of math-only RL training with progressively growing response length, i.e., Stage-1 (8K), Stage-2 (16K), and Stage-3 (24K), to develop a math-specialized RL model. We then apply code-only RL training to enhance model's coding capability. Lastly, we carry out a final stage of math-only RL to produce AceReason-Nemotron 1.1.

OpenCodeReasoning (Ahmad et al., 2025), and Llama-Nemotron (Bercovich et al., 2025). These models leverage large-scale math and code reasoning samples generated by DeepSeek-R1 (Guo et al., 2025) or QwQ (Qwen-Team, 2025). Some works describe the use of pretraining, supervised fine-tuning (SFT), and reinforcement learning (RL) in building strong reasoning models. However, a systematic study of the interplay and integration between SFT and RL is often lacking in technical reports (e.g., Yang et al., 2025; Xia et al., 2025; Seed et al., 2025).

## 3 METHOD

In this section, we present the details of the supervised fine-tuning and reinforcement learning processes used to train AceReason-Nemotron 1.1. The overall training pipeline is illustrated in Figure 1.

### 3.1 SUPERVISED FINE-TUNING

#### 3.1.1 PROMPT COLLECTION AND FILTERING

We collect math and code reasoning datasets from high-quality data sources. For math, we collect prompts from AceMath dataset (Liu et al., 2024c), NuminaMath (Li et al., 2024), and OpenMath-Reasoning (Moshkov et al., 2025). For coding, we collect prompts from TACO (Li et al., 2023), APPs (Hendrycks et al., 2021a), OpenCoder-Stage2 (Huang et al., 2024), and OpenCodeReasoning (Ahmad et al., 2025). We conduct the dataset deduplication to ensure that each prompt is unique. After that, we conduct data decontamination and filter the sample that has a 9-gram overlap with any test sample in math and coding benchmarks (Muennighoff et al., 2025). We use DeepSeek-R1 (Guo et al., 2025) to generate responses for the collected prompt set.

We aim for our SFT dataset to encompass a diverse range of challenging samples. Intuitively, longer model responses often correspond to more difficult questions. Based on this observation, we found that a large portion of the collected prompts are relatively simple, with many responses around or below 2,000 tokens in length. To achieve a better balance across difficulty levels, we randomly filtered out a subset of these simpler prompts and adjusted the proportions of other difficulty levels through additional random sampling. This resulted in a final dataset of 247K math prompts and 136K code prompts, totaling 383K prompts. Response token length distributions are in Appendix B.

#### 3.1.2 SCALING OF SFT DATA

We investigate how scaling the SFT dataset impacts model performance along two axes: (1) increasing the number of unique prompts, and (2) increasing the number responses per prompt. Expanding the set of prompts enriches the coverage of problem types and topics, while adding more responses per prompt allows the model to observe diverse reasoning paths for the same input.

To explore these scaling strategies, we construct seven SFT datasets (v1 through v7), each maintaining a similar distribution of response token lengths as shown in Appendix B. The dataset size scale increases from 36K samples in v1 to 2.2M samples in v7. In Section 4.4, Figure 2 presents the number of prompts and the corresponding average number of responses per prompt for each dataset.

For all datasets, SFT training is initialized from the base model Qwen2.5-Math-7B (Yang et al., 2024b), which is also the starting point for DeepSeek-R1-Distill-Qwen-7B (Guo et al., 2025). Since Qwen2.5-Math-7B only supports a context length of 4,096, we modify the `rope_theta` parameter from 10,000 to 1,000,000 enable support for a context length of 128K.

In Section 4.4, we investigate and address the following questions with respect to SFT:

1. How does scaling SFT data improve model performance on math and code benchmarks?

2. Is it more effective to scale the number of unique prompts or the number of responses per prompt? What is the best strategy in practice?

3. How does training for more epochs improve performance? What is the stopping criterion?

## 3.2 REINFORCEMENT LEARNING

### 3.2.1 OVERVIEW

We apply the stage-wise RL approach on math-only and code-only prompts in sequence, as described in Chen et al. (2025), to our SFT models. Specifically, we employ the GRPO algorithm (Shao et al., 2024) and strictly adhere to on-policy training by generating $G = 8$ or 16 rollouts $\{o_i\}_{i=1}^{G}$ for each question $q$ in a global batch of 128 prompts, followed by a single policy gradient update. The motivation for using on-policy training is to stabilize RL and prevent entropy collapse, as demonstrated in Chen et al. (2025). We utilize the token-level policy gradient loss, which assigns greater rewards to longer samples when the answer is correct and harsher penalties when it is incorrect. The intuition is that learning to generate longer samples plays a more critical role in enhancing reasoning capabilities. We remove the KL divergence term as well.

### 3.2.2 DATA CURATION

We utilize the high-quality math and code RL data from AceReason-Nemotron-1.0 (Chen et al., 2025). In particular, we find that the difficulty level of prompts and the accuracy of answers are the most critical factors for effective RL training. Prompts should be neither too easy nor too difficult, enabling the model to receive a balanced mix of positive and negative rewards across a group of rollouts. In all cases, answers must be as accurate as possible, as this is the only way for the model to receive meaningful signals during verification-based RL. Taken together, these findings highlight that the quality of RL data outweighs its quantity.

### 3.2.3 TRAINING PROCESS

We follow the math-only and code-only RL training curriculum introduced in AceReason-Nemotron-1.0 (Chen et al., 2025). We begin with math-only RL using a stage-wise response length extension from 8K to 16K, and then to 24K. Next, we apply code-only RL with response lengths of 24K and 32K. Finally, we conduct an additional round of math-only RL with a 32K response length budget. For each stage, we use the same RL training dataset as AceReason-Nemotron-1.0. The entire training process and key takeaways are summarized below.

1. **Math-only Stage-1 (8K)**: This initial stage with 8K response length budget serves as a warm-up phase for RL training. We use relatively simple questions sampled from our collected RL dataset for training. Most of these questions elicit responses from DeepSeek-R1 with token lengths predominantly between 2K and 4K.

2. **Math-only Stage-2 (16K)**: At this stage of training, we increase the proportion of more challenging questions compared to Stage 1. As a result, the model's average response length gradually increases, and we observe a substantial performance improvement.

3. **Math-only Stage-3 (24K)**: We filter out most of the simple questions and keep around 2500 hard ones for the training of this stage. Our model shows a significant performance improvement on math benchmarks in this stage.

4. **Code-only Stage-I (24K):** This stage marks the beginning of code RL training, which is initiated after math RL training to ensure greater stability.

5. **Code-only Stage-II (32K):** In this stage, we apply the epoch-wise filtering strategy as in AceReason-Nemotron-1.0. Specifically, we remove problems that can be fully solved by the previous epoch's checkpoint, i.e., problems for which every rollout passes all test cases.

6. **Math-only Stage-4 (32K):** As in the math-only Stage-3 (24K) setup, we filter out most of the simple questions—those that can be solved by every rollout—and retain only the challenging ones for training in this final stage.

In this work, we explore the following questions related to the RL process:

1. How does initializing RL from different SFT models affect final model performance? Does starting from a stronger SFT model lead to better overall performance?

2. How does the sampling temperature of policy LLM affect RL training, particularly in balancing the exploration–exploitation trade-off?

3. Is the Math-only Stage-1 (8K) truly necessary, given that it initially lowers benchmark performance? If the reasoning process compression from this stage is essential, how long should we train? Do we need to wait until benchmark accuracies fully recover?

4. During RL training, what is the best strategy when the final answer is not generated within a specified response length budget (e.g., 24K)? Should we assign a negative reward or mask out the entire sample (i.e., overlong filtering)?

5. Does RL still improve pass@K at large K when the SFT model is much stronger than DeepSeek-R1-Distill-Qwen used in AceReason-Nemotron-1.0.

## 4 EVALUATION

### 4.1 BENCHMARK

For math tasks, we evaluate our models on AIME24, AIME25, Math500 (Hendrycks et al., 2021b), as well as HMMT2025 Feb and BRUMO2025 from MathArena (Balunovic et al., 2025). For coding tasks, we evaluate our models on EvalPlus (Liu et al., 2023; 2024b), and LiveCodeBench v5 (2024/08/01–2025/02/01) and v6 (2025/02/01–2025/05/01) (Jain et al., 2024). Unless otherwise specified, all benchmarks use the default inference settings: a temperature of 0.6, top-p of 0.95, and a maximum sequence length of 32,768 (32K). Because reasoning models produce highly variable outputs when sampling is used, we report pass@1 performance averaged over $n$ runs (denoted as avg@n). For all reported numbers, we set $n = 64$ for AIME24, AIME25, HMMT2025 Feb, and BRUMO2025; $n = 8$ for LiveCodeBench V5 and V6; and $n = 4$ for MATH500 and EvalPlus. Note that a large n is crucial for obtaining a reliable metric, as the standard deviation of pass@1 decreases with $1/\sqrt{n}$—especially for benchmarks with a small number of problems. For AIME2024, avg@16, avg@32, and avg@64 yield stds of 1.8, 1.2, and 0.7, respectively.

### 4.2 BASELINES

Since our training begins with the base model Qwen2.5-Math-7B (Yang et al., 2024a), we primarily compare against state-of-the-art reasoning models built on either Qwen2.5 or Llama-3.1 (Grattafiori et al., 2024) of comparable parameter sizes to isolate the effects of pre-training and ensure a fair comparison. Our baselines include SFT models distilled from much larger frontier models (e.g., DeepSeek-R1), including Light-R1-7B (Wen et al., 2025), DeepSeek-R1-Distill-Qwen-7B (Guo et al., 2025), OpenMathReasoning-7B (Moshkov et al., 2025), and Llama-Nemotron-Nano-8B (Bercovich et al., 2025), as well as math-only RL-based models such as AReal-boba-RL-7B (RL Lab, 2025), Skywork-OR1-Math-7B (He et al., 2025a), and AceReason-Nemotron-1.0-7B (Chen et al., 2025). We also compare with state-of-the-art 7B code LLMs, including OlympicCoder-7B (HuggingFace, 2025) and OpenCodeReasoning-7B (Ahmad et al., 2025).

### 4.3 MAIN RESULTS

Table 1 shows the evaluation results on math and code benchmarks. Compared to other SFT models through distillation, our SFT model achieves slightly better results than Llama-Nemotron-Nano-8B-

| Models | AIME 2024 avg@64 | AIME 2025 avg@64 | MATH 500 avg@4 | HMMT 2025 avg@64 | BRUMO 2025 avg@64 | LiveCodeBench v5 avg@8 | LiveCodeBench v6 avg@8 | EvalPlus avg@4 |
|---|---|---|---|---|---|---|---|---|
| **SFT (distillation) based models:** | | | | | | | | |
| Light-R1-7B | 59.1 | 44.3 | 92.4$^\dagger$ | 27.6$^\dagger$ | 52.8$^\dagger$ | 40.6$^\dagger$ | 36.4$^\dagger$ | – |
| Llama-Nemotron-Nano-8B-v1 | 61.3 | 47.1 | 95.4 | – | – | 46.6 | 46.2$^\dagger$ | 81.2$^\dagger$ |
| OpenMath-Nemotron-7B | **74.8** | 61.2 | – | – | – | – | – | – |
| OpenCodeReasoning-Nemotron-7B | – | – | – | – | – | 51.3 | 46.1$^\dagger$ | 83.4$^\dagger$ |
| **RL based models:** | | | | | | | | |
| AReal-boba-RL-7B | 61.9 | 48.3 | 93.8$^\dagger$ | 29.4$^\dagger$ | 58.9$^\dagger$ | 34.3$^\dagger$ | – | – |
| Skywork-OR1-Math-7B | 69.8 | 52.3 | 94.4$^\dagger$ | 31.4$^\dagger$ | 60.6$^\dagger$ | 43.6 | – | – |
| Skywork-OR1-7B | 70.2 | 54.6 | 94.4 | 32.0$^\dagger$ | 59.7$^\dagger$ | 47.6 | 42.7$^\dagger$ | – |
| OlympicCoder-7B | – | – | – | – | – | 40.7 | 37.1$^\dagger$ | 79.8$^\dagger$ |
| MiMo-7B-RL | 68.2 | 55.4 | 95.8 | 35.7$^\dagger$ | 65.1$^\dagger$ | **57.8** | 49.3 | – |
| o3-mini (low) | 60.0 | 48.3 | 95.8 | 28.3 | 66.7$^\dagger$ | 60.9 | – | – |
| Magistral Small (24B) | 70.7 | 62.8 | 95.9 | – | – | 55.8 | 47.4 | – |
| DeepSeek-R1-Distill-Qwen-7B | 55.5 | 39.0$^\dagger$ | 92.8 | 26.3$^\dagger$ | 51.2$^\dagger$ | 37.6 | 34.1$^\dagger$ | 80.4$^\dagger$ |
| AceReason-Nemotron-1.0-7B | 69.0 | 53.6 | 94.1 | 33.9 | 62.2 | 51.8 | 44.1 | 84.6 |
| Our SFT-7B (starting point of RL) | 62.0 | 48.4 | 94.1 | 31.1 | 59.4 | 48.8 | 43.8 | 83.4 |
| AceReason-Nemotron-1.1-7B | 72.6 | **64.8** | 95.3 | **42.9** | **69.8** | 57.2 | **52.1** | **84.8** |

Table 1: Evaluation of reasoning models primarily based on Qwen2.5-Math 7B and Llama-3.1 8B to disentangle the impact of pretraining. We report pass@1 averaged over $n$ generations (avg@$n$) following the DeepSeek-R1 evaluation framework (same template, temperature=0.6, `top_p`=0.95, max response length=32768). **By default, we include official numbers from the model developers if they are available.** Otherwise, $^\dagger$we evaluate the model using the official template and same evaluation setting as above. Note that, unlike the base model Qwen2.5-Math, MiMo-7B-RL is developed from a base model pretrained with extensive synthetic reasoning data from advanced reasoning model (Xia et al., 2025).

v1, and achieves much better performance compared to Light-R1 and DeepSeek-R1-Distill-Qwen-7B. It is worth mentioning that, DeepSeek-R1-Distill-Qwen-7B is trained from Qwen2.5-Math-7B as our SFT model, suggesting the high quality of our collected SFT samples.

For AceReason-Nemotron-1.1-7B, we observe that the same AceReason RL training recipe (the same training method applied on the same training data) from Chen et al. (2025) substantially improves the performance of our strong SFT model, yielding absolute score gains of 10.6% on AIME24, 16.4% on AIME25, 8.4% on LiveCodeBench v5, and 8.3% on LiveCodeBench v6. As a result, AceReason-Nemotron-1.1-7B, demonstrates superior performance on math and code reasoning tasks, achieving the highest accuracy among 7B-scale models on AIME25 and LiveCodeBench v6—benchmarks that carry a lower risk of contamination compared to their earlier versions. As a reference, for AceReason-Nemotron-1.0-7B, the same RL training recipe improves its starting SFT model, DeepSeek-R1-Distill-Qwen-7B, 13.5% on AIME24, 14.6% on AIME25, 14.2% on LiveCodeBench v5, and 10.0% on LiveCodeBench v6. This demonstrates that a well-designed RL recipe can still largely boost the model's reasoning capability, even from a much stronger SFT model.

## 4.4 SCALING OF SFT DATA CONSISTENTLY IMPROVES PERFORMANCE

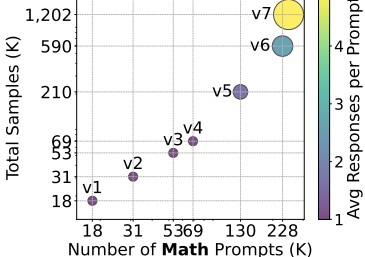
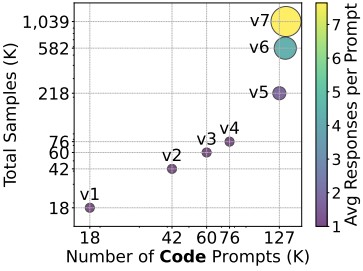

Figure 2: Log-scaled data statistics for the number of math and code prompts and the total number of samples. Circle size and color indicate the average responses for each prompt. Each SFT dataset consists of both math and code SFT samples.

Figure 2 presents the number of prompts, total samples, and average number of responses per prompt for each SFT dataset. Figure 3 demonstrates the benefits of scaling SFT datasets, from 18K math and 18K code samples in version v1 to 1.2M math and 1.0M code samples in version v7. We observe

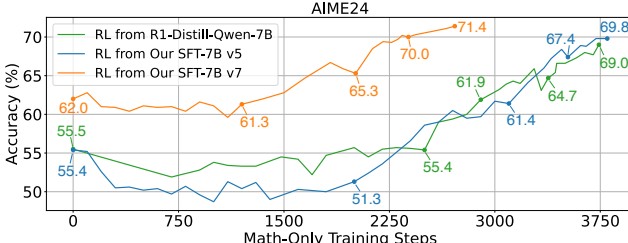

Figure 3: Accuracies on AIME24, AIME25, and LiveCodeBench V5 and V6 for different SFT datasets. For each SFT blend, the model is trained until the accuracy plateaus.

that both scaling strategies, namely increasing the diversity of prompts and adding more responses per prompt, significantly enhance model performance.

From the SFT dataset v1 to v4, we focus on expanding the number of unique prompts while limiting each prompt to a single response. Even modest number of unique prompt increases, such as adding 15K to 20K math or code samples, result in noticeable gains. For instance, expanding the math dataset by 16K samples from v3 to v4 leads to a 4% improvement in AIME24 and a 2% increase in AIME25. Starting with v5, we scale both the number of unique prompts and the number of responses per prompt. At this stage, the SFT v5 model matches the math performance of DeepSeek-R1-Distill-Qwen-7B and surpasses it in coding benchmarks. In our final dataset, v7, we maintain a similar number of prompts but further increase the number of responses per prompt. This additional scaling yields another performance boost, with AIME25 improving by 8% (from 41.3 to 49.3).

We examine which SFT data scaling factor more strongly influences performance and find that increasing the number of unique prompts has the greater impact. Detailed analyses are in Appendix C.

In addition, we observe that SFT performance continues to improve across multiple training epochs, plateauing around the 5th–6th epoch. Detailed analyses are provided in Appendix D.

## 4.5 RL ANALYSES

### 4.5.1 RL STARTING FROM DIFFERENT SFT MODELS

Figure 4: **Math-only** RL training starting from different SFT (distillation) models. The AIME24 accuracy at step-0 reflects the performance of the initial SFT checkpoints. The subsequent numbers in the figure show the final accuracy achieved at the end of each training stage: Math-Only Stage-1 (8K), Stage-2 (16K), Stage-3 (24K), and Stage-4 (32K).

Figure 4 presents RL experiments initialized from different SFT models, including two trained on our SFT datasets v5 and v7, as well as DeepSeek-R1-Distill-Qwen-7B. We generally observe significant performance gains at stage-2 (16K) and stage-3 (24K). The performance begins to plateau at stage-4 (32K) for the model initialized from SFT-7B v7, as further gains become more challenging on top of an already strong model. It is worth mentioning that these performance gains are accompanied by an increase in the average response token length at stage-2 and stage-3, suggesting that the model engaged in longer reasoning processes to tackle more difficult problems.

Interestingly, we observe that while some SFT models show substantial performance gaps (e.g., between SFT-7B v5 and v7), these differences become much smaller after applying RL training over more steps. For instance, the performance gap decreases from an initial 6.6% to 1.6% on AIME24. This underscores the potential of RL to effectively enhance model performance and bridge the gap between different starting points. Additionally, when RL is applied to SFT models with similar starting performance, such as SFT-7B v5 and DeepSeek-R1-Distill-Qwen-7B, the resulting performance on AIME24 reaches to a similar level. We put the results for AIME25 in Appendix E.

### 4.5.2 How training temperature affects the progress of RL

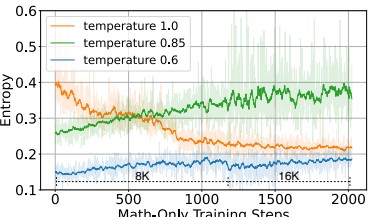

|  | AIME24 avg@64 | AIME25 avg@64 | LCB V5 avg@8 | LCB V6 avg@8 |
|---|---|---|---|---|
| **SFT Model (Step-0)** | | | | |
| Inference using temperature 0.6 | 62.0 | 48.4 | 48.8 | 43.8 |
| Inference using temperature 0.85 | 62.0 | 46.3 | 48.9 | 44.1 |
| Inference using temperature 1.0 | 61.2 | 45.4 | 48.5 | 43.6 |
| **Math-Only Stage-2 RL Models** | | | | |
| Trained with temperature 0.6 | 64.6 | 52.4 | 50.1 | 45.6 |
| Trained with temperature 0.85 | **67.6** | **56.8** | **52.1** | **47.1** |
| Trained with temperature 1.0 | 65.3 | 56.7 | 51.6 | 45.7 |

Figure 5: Left: Trajectories of temperature-adjusted entropy during RL training with different policy LLM temperature settings. Right: Impact of varying temperatures for inference and RL training. We observe that using a temperature of 0.6 for inference consistently yields better average results, and thus adopt 0.6 as the default inference temperature unless otherwise specified.

Figure 5 illustrates how different training temperature settings affect the trajectory of entropy during RL training and model performance after training. We make the following two observations:

- For a given model, the temperature in RL training should be carefully tuned—not set too low or too high. A low temperature (e.g., 0.6 in Figure 5) leads to over-exploitation and limited exploration, which can ultimately result in sub-optimal performance. In contrast, a high temperature (e.g., 1.0 in Figure 5) causes excessive exploration and low initial rewards, followed by a reduction in entropy and hindered learning progress.

- Through multiple trials, we observed a useful rule of thumb: setting the training temperature such that the temperature-adjusted entropy remains around 0.3 typically leads to effective RL training, as it strikes a good balance between exploration and exploitation.

Note that using a temperature of 0.6 for inference consistently yields better average results; therefore, we adopt 0.6 as the default inference temperature across experiments unless otherwise specified. However, with a training temperature of 0.6, RL begins with low entropy (around 0.15) and remains below 0.2 throughout the training, causing the policy LLM to favor exploitation over exploration. Such conservative learning behavior ultimately leads to sub-optimal performance.

With a training temperature of 1.0, entropy starts high (around 0.4), which gradually drops to approximately 0.22. This behavior may be attributed to the poor performance of the SFT model when sampling with a temperature of 1.0. As shown in Figure 5 (right table), we can see that using temperature 1.0 results in the poorest SFT models compared to 0.85 and 0.6. At the early RL stage, we also observe that using a temperature of 1.0 results in average rewards that are roughly 3–4% worse than those achieved with a temperature of 0.85. While a higher initial temperature encourages greater "exploration", a relatively low reward signal dampens this tendency, leading the model to favor "exploitation", as reflected in a sharper logits and a consequent drop in entropy.

In contrast, a moderate temperature of 0.85 starts with an entropy of approximately 0.26, which gradually increases to around 0.38 over the course of training. We hypothesize that the relatively higher rewards associated with this setting—also reflected in the table on the right (inference with 0.85 performs only marginally worse than 0.6, but significantly better than 1.0)—promote continued exploration, thereby driving the steady increase in entropy during training. This improved exploration–exploitation trade-off results in the highest benchmark performance.

### 4.5.3 At which stage should we apply overlong filtering?

During our curriculum RL training, model responses are sampled within a fixed response length budget (e.g., 8K or 16K tokens), and any outputs exceeding this limit are truncated. When a response surpasses this predefined length, a key training consideration emerges: should we mask the whole sample without assigning any reward (i.e., "w/ overlong filtering") or should such overlong outputs be penalized with a negative reward (i.e., "w/o overlong filtering")?

In previous studies, DAPO (Yu et al., 2025) shows that under a 16K token limit, penalizing truncated responses could confuse the model, as a well-reasoned answer might be unfairly penalized simply for its excessive length. Instead, applying overlong filtering helps stabilize training and improves

Figure 6: Ablation Studies on **Math-Only** RL training to assess the impact of overlong filtering. In both settings, Stage-1 starts with the same SFT model, and each subsequent stage begins with the same RL model from the previous stage trained under the best-performing setting (i.e., "w/ overlong filtering"). Notably, in the final stage (Stage-4), RL training without overlong filtering leads to superior performance. Evaluations are performed with a maximum sequence length of 32K. Results on AIME25 are in Appendix F.

|  | AIME24 avg@64 | AIME25 avg@64 | LCB V5 avg@8 | LCB V6 avg@8 |
|---|---|---|---|---|
| Math-Only Stage-4 RL w/ Overlong Filtering |  |  |  |  |
| Inference using 32K Maximum Length | 70.2 | 62.3 | 52.0 | 45.1 |
| Inference using 64K Maximum Length | 72.4 | 64.5 | 53.5 | 45.7 |
| Math-Only Stage-4 RL w/o Overlong Filtering |  |  |  |  |
| Inference using 32K Maximum Length | 71.4 | 63.5 | 53.5 | 48.0 |
| Inference using 64K Maximum Length | **73.0** | **64.8** | **54.5** | **48.7** |

Table 2: Comparisons of the effects of increasing the maximum output length to 64K, with and without applying the overlong filtering at the last stage of Math-Only RL.

performance. In contrast, Skywork-OR1 (He et al., 2025b) observes no clear performance benefit from overlong filtering, although their findings are limited to stage-1 (8K) RL training.

Compared to previous studies, we conduct a more systematic study for overlong filtering across all RL stages. Figure 6 illustrates the effects of "overlong filtering". Unlike Skywork-OR1, we observe a notable benefit from applying overlong filtering in Stage-1 (8K). This is largely due to the relatively short 8K token limit, where at the beginning of the training, approximately 30% of sample outputs exceed this boundary. Without overlong filtering, the negative rewards on these truncated samples will introduce significant noise into training. As the token limit increases, this noise diminishes—resulting in smaller gains from overlong filtering in Stage-2 (16K) and nearly equivalent performance in Stage-3 (24K). In Stage-4 (32K), RL training without overlong filtering outperforms the alternative. This is because removing overlong filtering makes the inference more token efficient and allows more concise generation within the 32K token budget.

Table 2 further evaluates models under a 64K maximum inference length. Interestingly, the model trained without overlong filtering—despite generating more concise outputs—is still able to outperform the model trained with overlong filtering, particularly on coding benchmarks. These findings contrast with the conclusions of DeepCoder (Luo et al., 2025a), which suggest that overlong filtering improves generalization to longer response lengths (e.g., 64K) during inference.

Additional analyses are provided in the Appendix due to space constraints. In Appendix G and H, we study the importance of stage-1 RL and its appropriate training duration. Appendix I investigates the impact of math-only RL on code reasoning performance. In Appendix J and K, we analyze how RL improves pass@k accuracies over the SFT model, particularly for large k. Finally, Appendix L explores how RL models perform on difficult problems compared to SFT models.

## 5 CONCLUSION

In this work, we study the training dynamics of supervised fine-tuning (SFT) and reinforcement learning (RL). We begin by analyzing SFT, examining the effects of scaling both the number of unique prompts and the number of responses per prompt. Our results show that both scaling strategies substantially improve the reasoning abilities of large language models (LLMs). Notably, performance consistently improves from the first to the fifth epoch during SFT, with gains plateauing around the fifth or sixth epoch—even when scaling the number of responses per prompt. We then conduct a systematic study of applying RL across different SFT models. Despite large initial performance differences among these models, RL training significantly reduces the performance gap. We observe that the first stage of RL training may not yield immediate improvements and can sometimes degrade performance. However, it plays a critical role in encouraging models to generate more concise reasoning, which proves beneficial in later stages of RL training. Interestingly, even

strong SFT models with robust coding abilities benefit substantially from math-only RL training. This leads to further gains in coding performance. Our final 7B model achieves state-of-the-art results among Qwen2.5-based 7B models, scoring 63.2% on AIME25 and 52.8% on LiveCodeBench V5—demonstrating strong performance on challenging math and code benchmarks.

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

## A INSTRUCTION FOR EVALUATION

### A.1 MATH

```
Please place your final answer inside \boxed{}.
```

### A.2 NO STARTER CODE (PYTHON)

```
Write Python code to solve the problem. Please place the solution code in
the following format:
```python
# Your solution code here
```
```

### A.3 HAS STARTER CODE (PYTHON)

```
Solve the problem starting with the provided function header.

Function header:
```
<starter_code>
```
Please place the solution code in the following format:
```python
# Your solution code here
```
```

## B SFT DATASET RESPONSE TOKEN LENGTH DISTRIBUTION

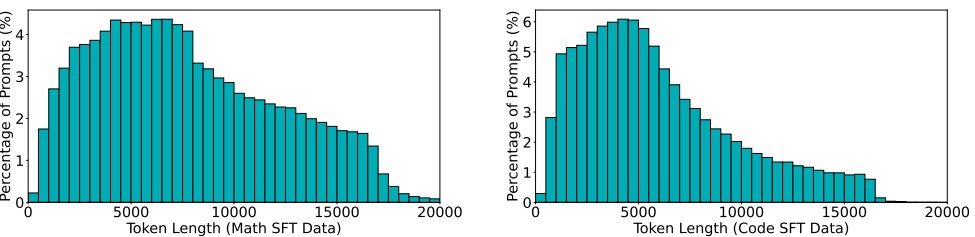

Figure 7: Response token length distributions for the math SFT dataset (left) and the code SFT dataset (right).

Figure 7 presents the response token length distributions for math and code prompts.

## C WHICH SFT DATA SCALING FACTOR HAS LARGER IMPACT

We further analyze which SFT data scaling factor contributes more significantly to performance improvements: increasing the number of unique prompts or increasing the number of responses per prompt. To this end, we perform a multiple linear regression analysis to model the relationship between overall accuracy ($z$) and the two independent variables: the number of unique prompts ($x$) and the number of responses per prompt ($y$), as described by the following equation:

$$z = a \cdot \log_2 x + b \cdot \log_2 y + c$$

We apply the least squares method to fit this model using seven data points (Figure 3) and estimate the regression coefficients $a$ and $b$, and bias $c$. Prior to fitting, $x$ and $y$ are tranformed to a log base-2 scale to reflect the exponential growth in total samples versus the linear trend in accuracy. Then, $x$

and $y$ are standardized (zero mean and unit variance) to ensure they are on the same scale. The dependent variable $z$ is defined as the average accuracy across AIME24, AIME25, and LiveCodeBench V5 and V6.

The resulting estimates are $a = 4.831$ and $b = 2.635$, with the coefficient of determination $R^2 = 0.989$, indicating a strong fit. The larger value of $a$ compared to $b$ suggests that increasing the number of unique prompts may have a greater impact on SFT model performance than increasing the number of responses per prompt. Given that when the number of unique prompts reaches a certain amount, increasing it becomes generally harder due to the difficulties of collecting more diverse data sources. In this case, increasing the number of responses for each prompt serves as a practical alternative to boost the performance of SFT model.

## D   SFT PERFORMANCE IMPROVES PROGRESSIVELY OVER EPOCHS

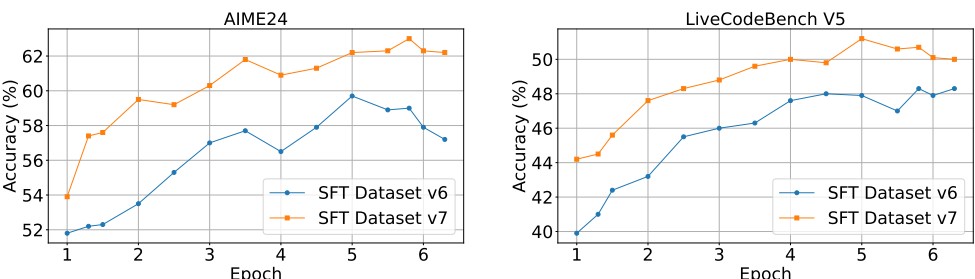

Figure 8: Accuracies over different epochs of training for SFT dataset v6 and v7.

Figure 8 illustrates the accuracies AIME24 and LiveCodeBench V5 over different epochs of training for SFT dataset v6 and v7. We observe that the model's performance gradually improves from the 1st to the 5th epoch, and begins to plateau around the 5th to 6th epoch. This pattern is consistent across different versions of the SFT dataset. This suggests that a certain degree of 'overfitting' may actually enhance test accuracy in long chain-of-thought (CoT) generation, likely due to *exposure bias* in autoregressive models. Additionally, we find that models trained on dataset v7 consistently outperform those trained on dataset v6 throughout training. While both versions share a similar set of prompts, dataset v7 contains nearly twice as many responses per prompt on average. Despite being exposed to the same prompts more frequently, models trained on v7 also reach a performance plateau around the fifth to sixth epoch. This suggests that increasing the number of responses per prompt in the SFT dataset also greatly benefits from multi-epoch training.

## E   RL TRAINING FROM DIFFERENT SFT MODELS ON AIME25

Consistent with the results shown in Figure 4, we observe from Figure 9 that the final performance gaps between RL models initialized from SFT-7B-v5 and SFT-7B-v7 narrows after training. However, a substantial gap remains when comparing RL training from SFT-7B-v7 to DeepSeek-R1-Distill-Qwen-7B. This suggests that when the initial SFT models differ significantly in quality, RL has limited ability to further close the performance gap.

## F   AT WHICH STAGE SHOULD WE APPLY OVERLONG FILTERING?

Extra results on AIME25 can be found in Figure 10, and the conclusion aligns with our findings on AIME24.

## G   IMPORTANCE OF STAGE-1 (8K)

We can find in Figure 4 that during Math-Only Stage-1 (8K), the model exhibits an initial drop in performance followed by a recovery. Nevertheless, its final performance at the end of Stage-1 may

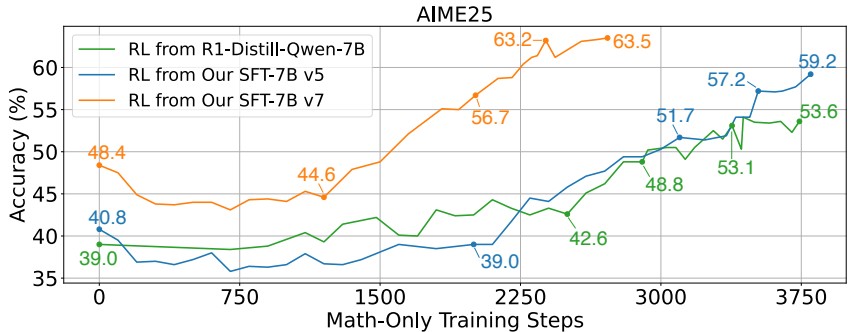

Figure 9: **Math-only** RL training starting from different SFT (distillation) models. The AIME25 accuracy at step-0 reflects the performance of the initial SFT checkpoints. The subsequent numbers in the figure show the final accuracy achieved at the end of each training stage: Math-Only Stage-1 (8K), Stage-2 (16K), Stage-3 (24K), and Stage-4 (32K).

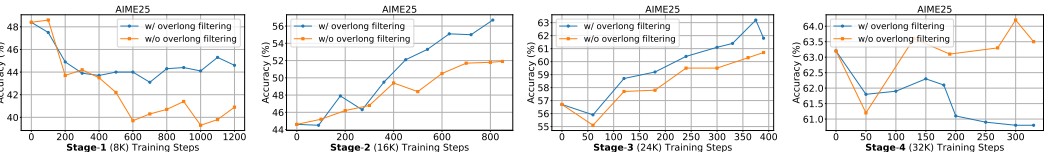

Figure 10: Ablation Studies on **Math-Only** RL training to assess the impact of overlong filtering. In both settings, Stage-1 starts with the same SFT model, and each subsequent stage begins with the same RL model from the previous stage trained under the best-performing setting (i.e., "w/ overlong filtering"). Notably, in the final stage (Stage-4), RL training without overlong filtering leads to superior performance. Evaluations are performed with a maximum sequence length of 32K.

remain below its initial level. Given the lack of clear improvement during Stage-1, a natural question arises: should Stage-1 be omitted?

To investigate this, we compare Stage-2 (16K) training initialized from the SFT checkpoint (omit Stage-1 (8K)) versus from the final step of Stage-1, as shown on the left side of Figure 11. We observe that although Stage-1 training causes an initial drop in AIME25 performance from 48.4 to 44.6, it enables more rapid and consistent improvement during Stage-2. In contrast, skipping Stage-1 results in slower gains and an eventual plateau at a lower performance level (56.7 vs. 51.8). As shown on the right side of Figure 11, we observe that the average response length sharply declines from over 5000 tokens to around 4000 tokens at approximately step 600 during stage-1 training. This reduction, occurring between steps 0 and 600, is attributed to the 8K token length limit and coincides with a period where the model performance drops. Although continued training after 600 steps improves performance, the average response length does not go up.

We conjecture that the 8K token constraint compels the reasoning model to develop a more concise thinking process in order to complete responses within the limit and successfully solve the tasks. One motivation for compressing the reasoning process is that the teacher-forced responses used during SFT are generated by DeepSeek-R1-671B, and may be too lengthy and complex for smaller models to effectively learn from or reproduce on their own. This reasoning compression becomes especially valuable during subsequent RL training, where the model maintains conciseness but benefits from a longer sequence length, allowing it to tackle more complex problems. In contrast, training an SFT model directly from stage-2 (with a 16K limit) bypasses this reasoning compression stage and thus fails to learn such capability.

## H  HOW LONG SHOULD WE TRAIN STAGE-1

From Appendix G, we learn that the stage-1 (8K) plays an important role in RL training. Building on this insight—and noting that model performance tends to recover and improve with continued

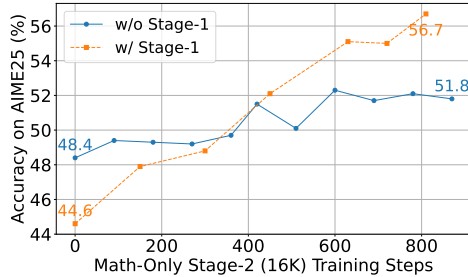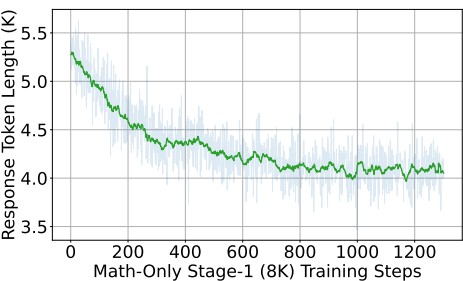

Figure 11: Left: Ablation study comparing models trained with and without Math-Only Stage-1. For "w/o Stage-1", the step-0 accuracy reflects the performance of the our SFT model on AIME25. In contrast, for "w/ Stage-1", the step-0 accuracy represents the final performance of Stage-1 RL initialized from the same SFT model. Right: Average response token length during Math-Only Stage-1 (8K) RL training.

| | AIME24 avg@64 | AIME25 avg@64 | LCB V5 avg@8 | LCB V6 avg@8 |
|---|---|---|---|---|
| Our SFT Model | 62.0 | 48.4 | 48.8 | 43.8 |
| Stage-1 1200 steps | 61.3(0.7↓) | 44.6(3.8↓) | 48.4 | 43.7 |
| + Stage-2 (16K) | 65.3(3.3↑) | 56.7(8.3↑) | 51.6 | 45.7 |
| Stage-1 1600 steps | 61.6(0.4↓) | 46.5(1.9↓) | 49.4 | 44.5 |
| + Stage-2 (16K) | 66.3(4.3↑) | 56.7(8.3↑) | 51.8 | 45.6 |
| Stage-1 2300 steps | 63.2(1.2↑) | 46.8(1.6↓) | 50.1 | 44.9 |
| + Stage-2 (16K) | 66.1(4.1↑) | 54.8(6.4↑) | 51.8 | 45.6 |

Table 3: Studies examining how varying the number of Stage-1 training steps (e.g., 8K) impacts subsequent RL training. The arrow indicates the accuracy comparison between our RL models and initial SFT model on math benchmarks.

stage-1 training—we investigate whether extending the duration of stage-1 training further enhances the subsequent RL stages.

Table 3 presents the results of this investigation, showing how varying the number of stage-1 training steps affects final performance at stage-2. We observe that continuing stage-1 training beyond 1200 steps yields slight additional improvements, with models gradually matching or even surpassing the original SFT model on math and coding benchmarks. However, these gains do not consistently translate to better outcomes at the final performance of stage-2. Specifically, models initialized from the stage-1 checkpoints at 1200, 1600, or 2300 steps result in comparable stage-2 performance. We also find that all stage-2 training plateau at similar training steps. This suggests that we could stop stage-1 training early and transition to stage-2, which leads to more substantial improvements.

## I   MATH-ONLY RL SIGNIFICANTLY IMPROVES CODE REASONING

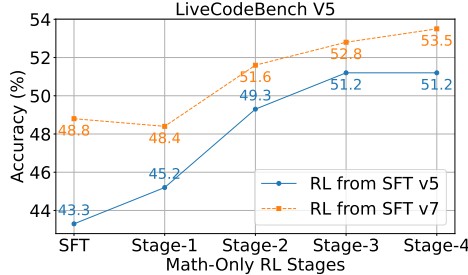

Figure 12: LiveCodeBench V5 accuracy over different **Math-Only** RL stages.

In AceReason-Nemotron-1.0 (Chen et al., 2025), we found that math-only RL significantly improves performance on code reasoning benchmarks. We reaffirm this finding using different SFT model as initialization, which are stronger than DeepSeek-R1-Distill-Qwen used in AceReason-Nemotron-1.0. To explore this further, we examine the impact of each math-only RL training stage on coding performance. Figure 12 reveals that the majority of the improvement stems from Stage-2, and Stage-1 can even potentially help improve coding capability for relatively weaker SFT model. Similar to our observations on math benchmarks, Stage-4 yields minor gains in coding performance.

Notably, the performance gap in coding narrows considerably over the course of RL training—from an initial 5.5% at the SFT starting point to just 1.6% by Stage-3. This trend mirrors the results observed on math benchmarks, further underscoring the powerfulness of RL training in enhancing model capabilities.

## J    RL IMPROVES UPON THE SFT MODEL IN TERMS OF PASS@K EVEN WHEN K IS LARGE

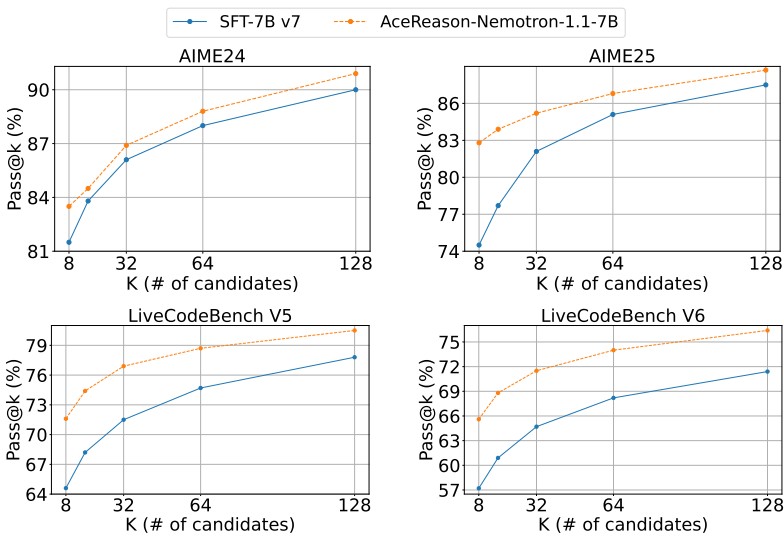

Figure 13: Comparison of pass@K scores between AceReason-Nemotron-1.1-7B and the SFT-7B v7 model it is trained from. To compute pass@K, we generate 256 outputs per sample for AIME24 and AIME25, and 128 outputs for LiveCodeBench V5 and V6. We then randomly select K outputs, and evaluate pass@K. This procedure is repeated 100 times, and the final pass@K score is computed by averaging the results across all repetitions.

The pass@K results from AceReason-Nemotron-1.0 (Chen et al., 2025) indicate that RL can consistently enhance pass@K with increasing K. As shown in Figure 13, our experiments on math and code benchmarks reveal that RL training consistently improves pass@k accuracy across the range of K = 8 to K = 128. These findings align with those of AceReason-Nemotron-1.0, despite our RL model being initialized from a considerably stronger SFT baseline (e.g., SFT-7B v7) compared to the DeepSeek-R1-Distill-Qwen-7B model used in their study.

In addition, we observe that for the math benchmark AIME25, the improvement from RL diminishes as K increases. For instance, the performance gain decreases from 8.3% at K = 8 to just 1.2% at K = 128. This is due to AIME's answer space is quite limited, consisting solely of positive integers with a maximum value in the hundreds. In contrast, the gains on coding benchmarks such as LiveCodeBench V5 and V6 remain substantial, with LiveCodeBench V6 still showing a 5% improvement at K = 128.

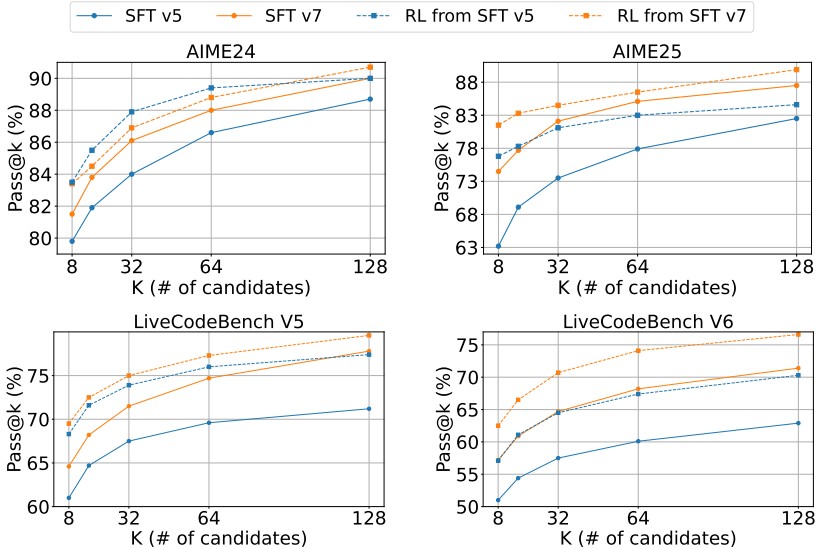

Figure 14: Pass@k results on AIME24, AIME25, LiveCodeBench V5, and V6, showcasing two SFT models and their subsequent **Math-Only** RL-trained versions. To compute pass@k, we generate 256 outputs per sample for AIME24 and AIME25, and 128 outputs for LiveCodeBench V5 and V6. We then randomly select k outputs, and evaluate pass@k. This procedure is repeated 100 times, and the final pass@k score is computed by averaging the results across all repetitions.

## K    PASS@K ACCURACY ON MATH-ONLY RL MODELS

In Figure 14, we demonstrate the pass@k results on two SFT models and their subsequent math-only RL-trained models. We find that the results are consistent with the observations in Appendix J. Similar to the pass@1 results, we observe that RL from a weaker SFT model (i.e., SFT v5) leads to greater improvements on pass@k, which further highlights the effectiveness of RL training. Interestingly, we find that even though the RL training is performed solely on math-specific prompts, the resulting models exhibit consistent gains on coding benchmarks (LiveCodeBench V5 and V6) as k increases.

## L    RL IMPROVES OVER STRONG SFT MODEL BY SOLVING HARD PROBLEMS

AceReason-Nemotron-1.0 (Chen et al., 2025) find that RL training is able to unlock a long tail of hard coding problems that the starting SFT model, DeepSeek-R1-Distill-Qwen-7B, fails to solve within 64 or even 1024 attempts. In particular, RL training notably boosts performance on problems where the SFT model achieves less than 20% accuracy. As shown in Figure 15, we observe that this finding still holds true, even though our initial SFT model is substantially stronger than the one used in AceReason-Nemotron-1.0 (Chen et al., 2025). Notably, on LiveCodeBench, we observe AceReason-Nemotron-1.1-7B is able to tackle a long tail of hard coding problems that the SFT model fails to solve within 128 attempts, leading to over ten additional problems solved on both LiveCodeBench V5 and V6.

## M    COMPUTATIONAL COST OF TRAINING

For SFT training, the total compute cost is 11,000 H100 GPU hours (458.3 GPU days).

For RL training, the total compute cost is 23,944 H100 GPU hours (997.7 GPU days), with the following breakdown:

- Math-only Stage-1: 2,773 H100 GPU hours (115.5 GPU days);
- Math-only Stage-2: 3,312 H100 GPU hours (138.0 GPU days);

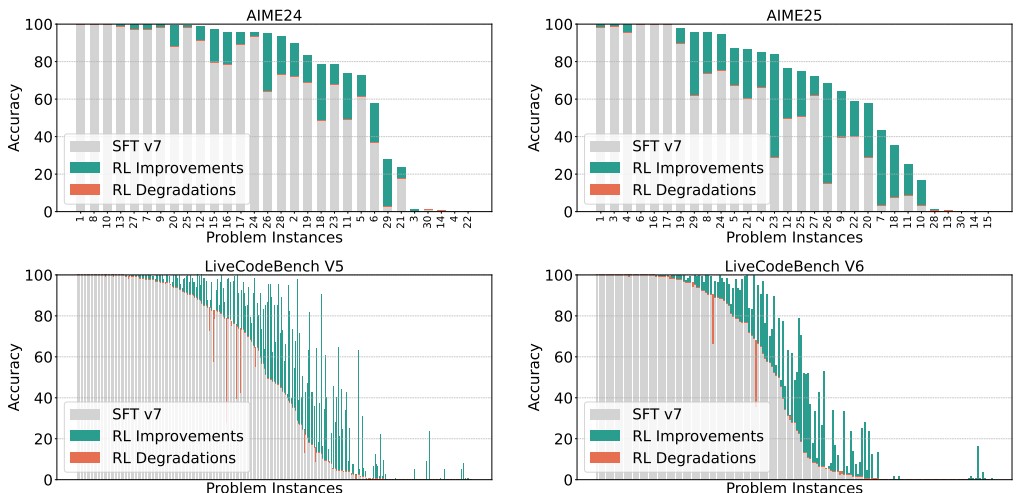

Figure 15: Comparison of problem-level solving rates between AceReason-Nemotron1.1-7B and the SFT-7B v7 model it is trained from. For each problem, accuracy is averaged over 256 outputs for AIME24 and AIME25, and over 128 outputs for LiveCodeBench V5 and V6.

- Math-only Stage-3: 5,084 H100 GPU hours (211.8 GPU days);
- Math-only Stage-4: 9,458 H100 GPU hours (394.1 GPU days);
- Code-only Stage-I: 1,875 H100 GPU hours (78.1 GPU days);
- Code-only Stage-II: 1,442 H100 GPU hours (60.1 GPU days)

