# OpenReview forum: "AceReason-Nemotron 1.1: Advancing Math and Code Reasoning through SFT and RL Synergy"
_ICLR.cc/2026/Conference — ICLR 2026 Poster_

### Official Review · Reviewer_hZwB · 2025-10-30

**Soundness:** 2
**Presentation:** 3
**Contribution:** 2
**Rating:** 4
**Confidence:** 4

**Summary:**

This paper studies the synergy between supervised fine-tuning and reinforcement learning in math and coding domain and and introduces AceReason-Nemotron-1.1-7B model that achieves high performance on math and coding problem. More specifically, they study the effects of scaling SFT data by increasing both prompt diversity and the number of responses per prompt. And they then apply a staged RL curriculum with increasing context length, analyzing how SFT strength, sampling temperature, and overlong response handling affect learning.

**Strengths:**

1. The paper collects a large amount of SFT and RL data and conducts thorough preprocessing. In addition, it performs extensive SFT and RL experiments along with many ablation studies, and evaluates the model across comprehensive math and coding benchmarks. Overall, the experiments are fairly thorough.


2. The final model achieves leading performance on multiple metrics compared to models of similar size, demonstrating the effectiveness of the proposed approach.

**Weaknesses:**

1. Although this paper features a very large workload and extensive experiments, I believe the generalization of some of its findings and conclusions is questionable. Specifically, all experiments are conducted on Qwen2.5-Math-7B and focus on math and coding generation tasks. From the model perspective, multiple works [1][2] have already pointed out the uniqueness of Qwen2.5-Math-7B, so the findings of this paper (e.g., sampling temperature choices) may not transfer to other models. From the task perspective, the experiments are primarily on math and coding. Would these conclusions still hold for more challenging tasks such as web agents or SWE-style agent tasks?

2. More broadly, the findings feel more like a report on tuning methods and hyperparameters for a particular model and task type, rather than offering deeper empirical and theoretical insights that generalize to different settings in the community. Therefore, I hope the authors can clearly distinguish which conclusions are generalizable and which only apply to the specific setup studied in the paper.

3. In the RL training phase, the authors divide the process into many stages, but no clear justification is provided. In Line 196, the authors cite AceReason-Nemotron-1.0 and follow a similar setup, yet do not explain the rationale behind these design choices in more detail.

[1] Zeng, Weihao, et al. "Simplerl-zoo: Investigating and taming zero reinforcement learning for open base models in the wild." arXiv preprint arXiv:2503.18892 (2025).

[2] Shao, Rulin, et al. "Spurious rewards: Rethinking training signals in rlvr." arXiv preprint arXiv:2506.10947 (2025).

**Questions:**

I have raised most of my concerns in the Weaknesses section, and I would like the authors to address them. Below are a few additional questions:

1. In Lines 396–398, the authors suggest keeping the entropy around 0.3. However, according to Figure 5, the entropy later deviates significantly from 0.3 and approaches 0.4. Could the authors clarify why this happens, and whether this suggestion is universal across settings (e.g. models, datasets)?

2. The paper conducts math and code training in separate stages. Have the authors considered joint training that mixes math and code signals instead of a staged approach?

---

> ### Author Response · Authors · 2025-11-22
> **Responses to Reviewer hZwB (Part 1)**
>
> Thank you so much for your valuable feedback. Let us address your questions below.
>
> - Q1: the generalization of some of its findings and conclusions is questionable. Specifically, all experiments are conducted on Qwen2.5-Math-7B and focus on math and coding generation tasks. From the model perspective, multiple works [1][2] have already pointed out the uniqueness of Qwen2.5-Math-7B, so the findings of this paper (e.g., sampling temperature choices) may not transfer to other models.
>
> Thanks for pointing out [1] and [2], we add them into the related work section.
>
> From [1], Qwen2.5-Math-7B base model differs from other base models in two key ways: (1) it has a relatively short 4K context window, and (2) it is a math-specialized model with strong performance in mathematical reasoning. To support longer contexts, we increase the RoPE θ parameter from 10,000 to 1,000,000 and run SFT to extend the usable context length to 16K. This allows the model to operate effectively in longer-context settings. Despite starting from this math-focused base model, our post-training pipeline yields substantial gains in both math (e.g., MATH500: 63.6 to 95.3, AIME24: 8.6 to 72.6) and code tasks, highlighting the effectiveness of our approach.
>
> [2] conducted a very interesting study on using different reward signals—including spurious rewards—and demonstrated the surprising effectiveness of RL on Qwen2.5-Math. However, their conclusions are primarily based on early training results (300 training steps) on MATH-500 (e.g., ~78.5%), which are substantially lower than the results reported in this paper (e.g., 95.3%). The performance gaps on AIME are even more pronounced (e.g., ~25% vs. 72.6% on AIME24).  Based on our experience, the uniqueness of Qwen2.5-Math-7B base is minimal in the large-scale training regime (>3000 steps) and high-performance levels.
>
> Importantly, we apply RL on top of SFT checkpoints rather than directly on pretrained models to reach high-performance regimes. Under this setting, the primary source of uniqueness lies in the SFT checkpoints themselves. For example, in Figures 4 and 9, we include comparisons that involve applying math RL to DeepSeek-R1-Distill-Qwen-7B, in addition to our SFT models. The results show that math RL also greatly enhances this model, and the final RL performance closely correlates with the SFT model’s initial performance.
>
> We believe many of our findings generalize beyond Qwen2.5-Math-7B. For instance, the importance of stage-1 RL training should extend to other SFT models, since models distilled from long-chain-of-thought data typically benefit from an initial compression stage. Moreover, our observations about overlong filtering are also model-agnostic: in early RL stages, when the model’s effective context length is still limited, overlong filtering helps reduce noise; in later stages, removing this filter encourages the model to solve harder problems within the available context. Regarding sampling temperature, our experiments indicate that setting the temperature-adjusted entropy around 0.3 provides a good balance between exploration and exploitation. While this value may vary slightly across models, our results show that keeping the entropy between 0.2 and 0.4 generally leads to strong RL training performance. Overall, these insights are not tied to any one model family and should extend to other models as well.
>
>
> - Q2: From the task perspective, the experiments are primarily on math and coding. Would these conclusions still hold for more challenging tasks such as web agents or SWE-style agent tasks?
>
> Our preliminary results indicate that continually applying RL on SWE-style GitHub issues–solving tasks improves performance on SWE Bench while maintaining math and code capabilities. We will do more studies on this in future work.
>
>
> - Q3: More broadly, the findings feel more like a report on tuning methods and hyperparameters for a particular model and task type, rather than offering deeper empirical and theoretical insights that generalize to different settings in the community. Therefore, I hope the authors can clearly distinguish which conclusions are generalizable and which only apply to the specific setup studied in the paper.
>
> Thanks for pointing this out. Our SFT and RL approaches, along with the experiments and analyses, are designed to develop advanced math and code models starting from any base model. The only component specifically tailored to our starting point, Qwen2.5-Math-7B base, is increasing the RoPE θ from 10,000 to 1,000,000 before SFT training, since the original model supports only a 4K context length and we aim to extend the context length. Aside from this adjustment, the insights and techniques in this paper are not tied to any particular model family and should generalize to other base models. We will add this discussion in our paper.

---

> ### Author Response · Authors · 2025-11-22
> **Responses to Reviewer hZwB (Part 2)**
>
> - Q4: In the RL training phase, the authors divide the process into many stages, but no clear justification is provided. In Line 196, the authors cite AceReason-Nemotron-1.0 and follow a similar setup, yet do not explain the rationale behind these design choices in more detail.
>
> Thank you for raising this point. As shown in Figure 11 (Appendix G), stage-1 (8K) RL training plays an important role by encouraging the model to adopt a more concise reasoning style. This conciseness becomes especially useful in later stages, where the model can retain token efficiency while benefiting from a longer sequence length. Moreover, gradually increasing the context length enables the model to tackle progressively harder tasks that inherently require longer reasoning. While difficult math and code problems demand extended reasoning processes, exposing the model to long-sequence sampling too early leads to unstable optimization and weak reward signals. We will provide an additional explanation of this rationale in the paper.
>
>
> - Q5: In Lines 396–398, the authors suggest keeping the entropy around 0.3. However, according to Figure 5, the entropy later deviates significantly from 0.3 and approaches 0.4. Could the authors clarify why this happens, and whether this suggestion is universal across settings (e.g. models, datasets)?
>
> Figure 5 shows that using a temperature of 0.85 (corresponding to an initial entropy of roughly 0.26) achieves noticeably higher rewards than using a temperature of 1.0. This trend is further confirmed by the right table in Figure 5, where inference at 0.85 temperature substantially outperforms inference at 1.0 temperature. Higher rewards at temperature 0.85 encourage continued exploration, which in turn drives the steady rise in entropy throughout training. In contrast, the lower initial rewards at temperature 1.0 cause entropy to decline during RL training. We believe that this 0.3 entropy suggestion can be applied for other models, since entropy in sampling is a property of the probability distribution produced by the model, not the architecture.
>
>
> - Q6: The paper conducts math and code training in separate stages. Have the authors considered joint training that mixes math and code signals instead of a staged approach?
>
> In this work, we adopt a stage-wise RL approach for several reasons. First, unlike SFT, RL training on a specific domain (e.g., math) does not negatively impact other domains (e.g., code); in fact, math RL can even enhance coding performance by improving the model’s reasoning capabilities. Second, combining math and code in RL training is more challenging, as obtaining reward signals differs in timing (coding tasks typically require longer to produce rewards) which can slow down training. We plan to explore mixed math and code RL training in future work.

---

### Official Review · Reviewer_3y3R · 2025-11-01

**Soundness:** 3
**Presentation:** 3
**Contribution:** 2
**Rating:** 6
**Confidence:** 4

**Summary:**

This work presents a comprehensive study of SFT and RL training on a 7B reasoning model. The authors investigate different aspects, including the impact of SFT model on RL performance, the sampling temperature and overlong penalty during RL training. The whole RL training recipe starts from math training to code training, with a short-to-long generation length. The resulted model achieves competitive results on both math and code benchmarks.

**Strengths:**

1. The paper presents a comprehensive SFT&RL recipe for a 7B reasoning models.
2. Ablation studies from different aspects are conducted, making this work as a solid technical work and providing valuable empirical insights.
3. The resulted model achieves competitive performance on both math and code benchmarks.

**Weaknesses:**

1. The question "Is theMath-onlyStage-1(8K) trulynecessary" is not answered in the submission.

**Questions:**

1. Why does the authors choose to apply RL training on math and code in seperate stages, instead of apply RL training on math and code data simultaneously?

---

> ### Author Response · Authors · 2025-11-22
> **Responses to Reviewer 3y3R**
>
> Thank you so much for your valuable feedback. Let us address your questions below.
>
>
> - Q1: The question "Is the Math-only Stage-1(8K) truly necessary" is not answered in the submission.
>
> Due to the page limit, we put the analysis for this research question in Appendix G. We observe that although Stage-1 (8K) training causes an initial drop in AIME25 performance from 48.4 to 44.6, it enables more rapid and consistent improvement during Stage-2 (16K).
>
> We also find that the response length sharply declines from over 5000 tokens to around 4000 tokens  in Stage-1 (8K) training. We conjecture that the 8K token constraint compels the reasoning model to develop a more concise thinking process in order to complete responses within the limit and successfully solve the tasks. This reasoning compression becomes especially valuable during subsequent RL training, where the model maintains conciseness but benefits from a longer sequence length, allowing it to tackle more complex problems.
>
>
>
> - Q2: Why does the authors choose to apply RL training on math and code in separate stages, instead of apply RL training on math and code data simultaneously?
>
> We find that RL training on a specific domain (e.g., math) does not negatively impact other domains (e.g., code); in fact, math RL can even enhance coding performance by improving the model’s reasoning capabilities. Moreover, combining math and code in RL training is more challenging, as obtaining reward signals differs in timing (coding tasks typically require longer to produce rewards) which can slow down training. We will explore mixed math and code RL training in future work.

---

### Official Review · Reviewer_dEtu · 2025-11-01

**Soundness:** 3
**Presentation:** 3
**Contribution:** 4
**Rating:** 8
**Confidence:** 4

**Summary:**

This paper conducts a systematic analysis of the alternating interplay and integration between SFT (Supervised Fine-Tuning) and RL (Reinforcement Learning) training. It explores how expanding the SFT dataset improves initialization — including the number of prompts, the number of responses per prompt, and the number of training epochs. The authors also investigate the impact of using different SFT models for initializing RL, as well as how varying the temperature during RL training affects entropy and accuracy. In addition, they examine whether filtering overlong samples during RL affects the outcomes. Through detailed experiments, the authors study all these questions and demonstrate that SFT based on Qwen2.5-7B surpasses DeepSeek-R1-Distill-Qwen in performance. Furthermore, they confirm that stage-wise RL can achieve further improvements on stronger SFT models.

**Strengths:**

1. Compared with most technical reports that provide only brief descriptions of SFT and RL details, this paper conducts a systematic study on the interplay and integration between SFT and RL using extensive resources. This work is highly significant for understanding the training dynamics of RL initialized from SFT in the research community.

2. The authors analyze SFT data in detail — including the number of prompts, the number of responses per prompt, and their effects on SFT performance — across both Math and Code domains. They also study how different SFT models influence RL training dynamics, offering valuable insights. For example, scaling the number of prompts yields more significant improvements, and stronger SFT models lead to higher RL starting points, though the performance gap narrows as RL training progresses.

3. The authors analyze how different temperatures in RL affect entropy and performance, experimentally illustrating healthy entropy dynamics and corresponding temperature settings during RL training.

4. They further study the impact of overlong sample filtering, providing detailed analyses under various maximum-length settings.

**Weaknesses:**

The experiments are mainly conducted on Qwen2.5-based 7B models. It is unclear whether the authors tested larger models, such as 32B, to verify the generalizability of their conclusions. However, given the large-scale data and the computational resources required, the absence of such experiments is understandable.

**Questions:**

See in Weakness

---

> ### Author Response · Authors · 2025-11-22
> **Responses to Reviewer dEtu**
>
> Thank you so much for your valuable feedback. Let us address your questions below.
>
>
> - Q1: The experiments are mainly conducted on Qwen2.5-based 7B models. It is unclear whether the authors tested larger models, such as 32B, to verify the generalizability of their conclusions. However, given the large-scale data and the computational resources required, the absence of such experiments is understandable.
>
> Thank you for pointing this out. We didn’t test larger models due to the limit of computational resources. However, in Figures 4 and 9, we add comparisons that involve applying math RL to DeepSeek-R1-Distill-Qwen-7B. The results show that math RL also greatly enhances this model.

---

> > ### Comment · Reviewer_dEtu · 2025-11-25
> >
> > Thanks for your response. The absence of large scale experiments is understandable. I also keep my positive score for accept. Good luck!

---

### Official Review · Reviewer_aouA · 2025-11-01

**Soundness:** 4
**Presentation:** 3
**Contribution:** 4
**Rating:** 8
**Confidence:** 4

**Summary:**

The paper studies how supervised fine-tuning (SFT) and reinforcement learning (RL) interact to improve long chain-of-thought reasoning for math and code. It scales SFT along two axes with more unique prompts and more responses per promptand analyzes epoch effects. It then applies a stage-wise, strictly on-policy GRPO RL curriculum (math stages at 8K -> 16K-> 24K-> 32K tokens, interleaved with code RL at 24K and 32K) and examines starting-point sensitivity, sampling temperature, and handling of over-length trajectories. The resulting 7B model, AceReason-Nemotron-1.1, built on Qwen2.5-Math-7B, reports improved results on AIME24/25, MATH500, HMMT2025, BRUMO2025, EvalPlus, and LiveCodeBench v5/v6, with analyses of temperature-adjusted entropy and overlong filtering.

**Strengths:**

- This is a well-executed empirical paper that systematically probes how supervised fine-tuning and reinforcement learning interact for long chain-of-thought reasoning, with clear takeaways (e.g., entropy/temperature rule-of-thumb, stage-wise curriculum, and overlong-filtering policy). The paper offers generalizable training guidance.

- The empirical program is rigorous and targeted: clearly defined questions, extensive ablations, and transparent evaluation settings.

- The model is evaluated on a wide range of mathematical and coding benchmarks, using repeated sampling avg@n for variance reduction and consistent default decoding across all models.

**Weaknesses:**

- Please report GPU-days per stage to contextualize the recipe’s practicality.
- Adding one/more non-Qwen-7B model would validate generality.
- The authors did not release code, which may limit reproducibility.
- In Figure 4, why is the “final accuracy achieved at the end of each training stage” for different models not shown at the same step? Does this mean that different models require a different number of training steps in a given stage?
- Regarding the results in Figure 4, have you conducted experiments where RL is applied directly to the base model (skipping SFT)? Would this lead to similar conclusions? Additionally, the reviewer is also curious whether applying RL directly on an existing instruct model would produce more interesting comparison results.
- During different stages of RL training, is there any overlap between the samples used in each stage? How many samples were used in total for each stage?
- How is the number of training steps for each RL stage determined? What are the exact numbers?
- In the SFT experiments on the number of responses per prompt, does the diversity of the responses per prompt have a significant impact? When sampling multiple responses, how do the authors ensure diversity?

**Questions:**

See Weaknesses

---

> ### Author Response · Authors · 2025-11-22
> **Responses to Reviewer aouA**
>
> Thank you so much for your valuable feedback. Let us address your questions below.
>
>
> - Q1: Please report GPU-days per stage to contextualize the recipe’s practicality.
>
> Thanks for raising this up. We report the GPU-days for our best model below.
>
> For SFT training, the total compute cost is 11,000 H100 GPU hours (458.3 GPU days).
>
> For RL training, the total compute cost is 23,944 H100 GPU hours (997.7 GPU days), with the following breakdown:
>
> Math-only Stage-1: 2,773 H100 GPU hours (115.5 GPU days);
>
> Math-only Stage-2: 3,312 H100 GPU hours (138.0 GPU days);
>
> Math-only Stage-3: 5,084 H100 GPU hours (211.8 GPU days);
>
> Math-only Stage-4: 9,458 H100 GPU hours (394.1 GPU days);
>
> Code-only Stage-I: 1,875 H100 GPU hours (78.1 GPU days);
>
> Code-only Stage-II: 1,442 H100 GPU hours (60.1 GPU days)
>
> We will include these GPU-day details in the paper.
>
>
> - Q2: Adding one/more non-Qwen-7B model would validate generality.
>
> In Figures 4 and 9, we have comparisons that involve applying math RL to DeepSeek-R1-Distill-Qwen-7B. The results show that math RL also greatly enhances this model, and the final RL performance closely correlates with the SFT model’s initial performance.
> We will conduct experiments starting more base models in future work.
>
>
> - Q3: The authors did not release code, which may limit reproducibility.
>
> Our RL training follows the VERL framework (https://github.com/volcengine/verl), with additional training details provided in Section 3.2.1. We will further expand on these details to ensure full reproducibility.
>
>
> - Q4: In Figure 4, why is the “final accuracy achieved at the end of each training stage” for different models not shown at the same step? Does this mean that different models require a different number of training steps in a given stage?
>
> Thanks for pointing this out. Our goal is to apply RL to each SFT model and maximize its performance for a more fair comparison. Stopping RL at the same step for all models could introduce larger variance, as different models may reach their optimal performance at different points. Therefore, for each RL stage, we continue training until performance saturates before proceeding to the next stage. The most notable difference in training steps occurs during Stage-1 (8K) training, where we observe an initial drop in performance; in this case, RL continues until performance recovers or stabilizes. We will include this clarification in the paper.
>
>
>
> - Q5: Regarding the results in Figure 4, have you conducted experiments where RL is applied directly to the base model (skipping SFT)? Would this lead to similar conclusions? Additionally, whether applying RL directly on an existing instruct model would produce more interesting comparison results.
>
> We have not yet explored applying RL directly to the base model. We will put them in the future work. In Figures 4 and 9, we compare the effects of applying math RL to an existing instruction-tuned model (SFT model), DeepSeek-R1-Distill-Qwen-7B. The results indicate that math RL significantly improves this model as well.
>
>
>
> - Q6: During different stages of RL training, is there any overlap between the samples used in each stage? How many samples were used in total for each stage?
>
> There is some overlap between stages, with the proportion of difficult problems gradually increasing. For math, we use 49k samples in Stage-1, 38k in Stage-2, and 2.2k in both Stage-3 and Stage-4. For coding, we use 8k samples in Stage-I and 6k in Stage-II. We will include them in the paper.
>
>
> - Q7: How is the number of training steps for each RL stage determined? What are the exact numbers?
>
> For each RL stage, we continue training until the performance plateaus before moving to the next stage. This ensures that the model reaches its best possible performance. Below are the exact RL training steps used for SFT v7. For math-only RL, we train for 1200 steps in Stage-1, 810 in Stage-2, 260 in Stage-3, and 280 in Stage-4. For code-only RL, we train 140 steps for Stage-I and 75 steps for Stage-II.
>
>
>
> - Q8: In the SFT experiments on the number of responses per prompt, does the diversity of the responses per prompt have a significant impact? When sampling multiple responses, how do the authors ensure diversity?
>
> Thanks for pointing this out. In our SFT experiments, we did not analyze how response diversity varies across prompts. We did observe that answers to the same prompt can differ by roughly 2K tokens in length. Exploring how response diversity affects SFT performance would be an interesting direction for future work.

---

> > ### Comment · Reviewer_aouA · 2025-11-25
> >
> > Thanks for your response. I keep my positive score for accept. Good luck!

---

### Meta-Review · Area_Chair_Su6K · 2026-01-06

**Summary:**

The reviewers rated this submission as a well-executed empirical study on how SFT and GRPO  interact for chain-of-thought reasoning in math and code. The original ratings are 8, 8, 6, 4.

The major concerns are about the generality beyond Qwen2.5-Math-7B  and math & code tasks, as well as the experimental design of ints  staged RL recipe. These concerns are valid but do not form the basis to reject the original submission of this work. That being said, together with its relatively high ratings, this work is recommended with an acceptance.

**Reviewer Concerns:**

Most concerns were light given the high ratings, and addressed by the authors' rebuttal, including training cost, RL training pipeline, mix of code and math.

The questions that are not fully addressed are difficult to face for most submissions, including the generalization of the studies / observations to larger size of language models, and the generalization of code and moth.

**Reviewer Scores:**

It is most likely that all four reviewers would not downgrade this work.
aouA: 8
dEtu: 8
3y3R: 6
hZwB: 4

---

### Decision · Program_Chairs · 2026-01-26

Accept (Poster)